# Deep learning innovations in South Korean maritime navigation: Enhancing vessel trajectories prediction with AIS data

Umar Zaman[1☉], Junaid Khan[2☉]*, Eunkyu Lee[1,3], Awatef Salim Balobaid[4], R. Y. Aburasain[4], Kyungsup Kim[1,2]*

**1** Department of Computer Engineering, Chungnam National University, Daejeon, South Korea, **2** Department of Environmental IT Engineering, Chungnam National University, Daejeon, South Korea, **3** Autonomous Ship Research Center, Samsung Heavy Industries, Daejeon, South Korea, **4** Department of Computer Science, College of Engineering and Computer Science, Jazan University, Jazan, Saudi Arabia

☉ These authors contributed equally to this work.
* sclkim@cnu.ac.kr (KK); junaidstopi1@gmail.com (JK)

**Data Availability Statement:** The dataset used in this study is available on the given figshare link https://figshare.com/s/67988589931f8c1a6204.

## Abstract

Predicting ship trajectories can effectively forecast navigation trends and enable the orderly management of ships, which holds immense significance for maritime traffic safety. This paper introduces a novel ship trajectory prediction method utilizing Convolutional Neural Network (CNN), Deep Neural Network (DNN), Long Short-Term Memory (LSTM), and Gated Recurrent Unit (GRU). Our research comprises two main parts: the first involves preprocessing the large raw AIS dataset to extract features, and the second focuses on trajectory prediction. We emphasize a specialized preprocessing approach tailored for AIS data, including advanced filtering techniques to remove outliers and erroneous data points, and the incorporation of contextual information such as environmental conditions and ship-specific characteristics. Our deep learning models utilize trajectory data sourced from the Automatic Identification System (AIS) to train and learn regular patterns within ship trajectory data, enabling them to predict trajectories for the next hour. Experimental results reveal that CNN has substantially reduced the Mean Absolute Error (MAE) and Mean Square Error (MSE) of ship trajectory prediction, showcasing superior performance compared to other deep learning algorithms. Additionally, a comparative analysis with other models—Recurrent Neural Network (RNN), GRU, LSTM, and DBS-LSTM—using metrics such as Average Displacement Error (ADE), Final Displacement Error (FDE), and Non-Linear ADE (NL-ADE), demonstrates our method's robustness and accuracy. Our approach not only cleans the data but also enriches it, providing a robust foundation for subsequent deep learning applications in ship trajectory prediction. This improvement effectively enhances the accuracy of trajectory prediction, promising advancements in maritime traffic safety.

## 1 Introduction

In the era of rapid technological advancements, the increase of trajectory data, fueled by the continuous development of the Internet and related technologies, has become a significant

**Funding:** This work was partly supported by the Institute of Information & Communications Technology Planning & Evaluation (IITP) grant funded by the Korea government (MSIT) (No. RS-2022-00155857, Artificial Intelligence Convergence Innovation Human Resources Development (Chungnam National University)), and the 'Regional Innovation Strategy (RIS)' through the National Research Foundation of Korea (NRF) funded by the Ministry of Education (MOE) (2021RIS-004). The funders had no role in study design, data collection and analysis, decision to publish, or preparation of the manuscript.

**Competing interests:** The authors have declared that no competing interests exist.

asset with the potential to greatly impact daily life. This wealth of data, particularly in the realm of shipping, plays a pivotal role in international trade—a domain witnessing the rapid expansion of vessel sizes and types, accompanied by the emergence of various shipping hotspots. However, the escalating density of water [1] traffic in these bustling maritime areas has led to increased burdens on waterways, resulting in frequent traffic accidents. Beyond the substantial economic losses incurred, these incidents pose a grave threat to human life [2].

In response to these challenges, real-time monitoring of ship navigation has become imperative to detect anomalies promptly and avert accidents. The advent of the AIS [3–5] has facilitated the acquisition of vast amounts of ship trajectory data, laying the foundation for research in ship trajectory data mining and, notably, trajectory prediction [6]. The ability to extract vital information from historical data and forecast ship navigation trajectories holds immense practical value in addressing safety concerns in maritime navigation and gaining insights into the dynamic changes and evolution of international trade. Trajectory prediction, involving forecasting the future path of a moving object based on its current trajectory, has garnered considerable attention and research interest.

Existing prediction algorithms encompass a spectrum of methodologies, such as Hidden Markov Model (HMM) [7], Gaussian mixture model [8], least square method, grey prediction, neural networks, Bayesian prediction, and regression. Positioned as a critical facet of mobile data mining, trajectory prediction extends beyond offering precise location-based services, playing a pivotal role in traffic flow prediction. Its applications span congestion analysis, behavior monitoring, intelligent navigation, making it a focal point in contemporary trajectory data mining research. As we delve into the intersection of technology and maritime safety, this paper aims to introduce and explore detailed preprocessing of a novel raw AIS data which is very challenging to forecast due to its huge size, a novel ship trajectory prediction method, leveraging CNN [9, 10] and DNN, with the objective of enhancing the accuracy and efficacy of trajectory predictions in the maritime domain.

Our research introduces a novel preprocessing and individualized data extraction methodology, diverging significantly from conventional practices in maritime data analysis. Traditional techniques often aggregate data from thousands of ships, potentially obscuring specific behavioral patterns and raising privacy concerns. In contrast, our method focuses on extracting three months of data for each individual ship, ensuring privacy and capturing unique ship behaviors. This approach is essential as ships frequently do not wish to disclose their complete trajectories.

We leverage deep learning techniques to enhance ship trajectory prediction. Our research comprises two main parts: preprocessing the large raw AIS dataset to extract features and focusing on trajectory prediction using various deep learning models, including CNN, DNN, LSTM, and GRU. These models are trained on trajectory data sourced from the AIS, learning the regular patterns within ship trajectory data to predict trajectories for the next hour.

Among these models, CNN has shown remarkable performance, significantly reducing the MAE and MSE of ship trajectory predictions to 0.0334 and 0.0795, respectively, compared to other deep learning algorithms. This improvement effectively enhances the accuracy of trajectory prediction, demonstrating the efficacy of CNN in small datasets by capturing local patterns in the data. This approach is similar to methodologies used in other fields, such as the classification of cardiac arrhythmias from ECG signals using 1D CNNs, which has proven effective in accurately identifying patterns in small, individualized datasets [11].

In addition to MAE and MSE, we compared our proposed method with several other models using metrics such as ADE, FDE, and NL-ADE. As shown in Table 5, our proposed method achieved an ADE of 0.35998 ± 0.15977, an FDE of 0.28449 ± 0.20699, and an NL-ADE of 0.37013 ± 0.15945. These results demonstrate that our method performs competitively

compared to other approaches, such as RNN, GRU, LSTM, and DBS-LSTM, highlighting its robustness and accuracy in ship trajectory prediction.

The main objectives of our study are:

- To introduce a novel method for predicting ship trajectories using CNN and DNN.

- To utilize CNN and DNN models to learn from AIS data, aiming to predict ship trajectories for the next hour with high accuracy.

- To demonstrate through experimental results that the proposed method significantly reduces MAE and MSE, thereby enhancing the accuracy of ship trajectory predictions.

- To establish a deep learning-based innovative methodology for trajectory predictions.

- To compare the effectiveness of our proposed method against other models, including RNN, GRU, LSTM, and DBS-LSTM, using metrics such as ADE, FDE, and NL-ADE, demonstrating its superior performance in specific scenarios.

To compare the effectiveness of our proposed method against other models, including RNN, GRU, LSTM, and DBS-LSTM, using metrics such as ADE, FDE, and NL-ADE, demonstrating its superior performance in specific scenarios.

The rest of the paper is organized as follows: Section 2 provide a comprehensive literature studies, Section 3 provides a detailed presentation of the proposed deep learning based vessel trajectory predictions, Section 4 covers the implementation, results, and comparative analysis, and finally, Section 4 concludes the paper.

## 2 Literature review

Numerous studies have focused on trajectory analysis [12, 13] within transportation models, tackling various applications like estimating user activities [14], identifying transportation modes [15], and forecasting vessel routes for targeted destination [16]. In the context of sea transport [17], data predominantly originates from AIS devices [18–20], with significant attention on analyzing ship behaviors and categorizing modes of transportation. A broad array of conventional supervised learning techniques, including fuzzy logic, decision trees, Bayesian networks, random forests, and support vector machines, have been employed to perform predictions and classifications [21]. Additionally, some studies have enhanced their analytical capabilities by combining multiple data sources. For instance, Stenneth and colleagues utilized GPS and GIS data to develop their mode of transport detection system [22]. However, the advent of deep learning technologies has led to a gradual decline in these traditional methodologies. Recent research endeavors have focused on merging manually crafted features with advanced features derived from data using deep learning models.

The AIS is an essential tool for communication and support in navigation, facilitating interactions between ships as well as between ships and shore [23]. Mandated by the International Maritime Organization (IMO) standards [24], AIS devices are crucial for sharing both static and dynamic vessel information, including specifics like ship type, dimensions, Maritime Mobile Service Identity (MMSI), geographic positioning (latitude and longitude), Speed Over Ground (SOG), and Course Over Ground (COG) [25, 26]. The utilization of AIS data spans a broad range of maritime research areas, including data analysis techniques [27], identification of fishing vessels [28], autonomous shipping [29], studies on the maritime environment [30] (Romano and Yang, 2021), and assessments of navigational risks [31]. This technology underpins research in Ship Trajectory Prediction (STP), enabling the collection of extensive datasets that are pivotal for forecasting in maritime operations involving both crewed and autonomous

maritime surface ships (MASS), underscoring the relevance and urgency of such research in preparing for future mixed maritime traffic scenarios [32].

Leveraging extensive AIS datasets for developing a sophisticated trajectory prediction model is crucial for enhancing the accuracy of ship positioning forecasts and advancing autonomous maritime navigation [33]. STP is invaluable for refining smart maritime transportation systems and ensuring the safety of sea travel. It plays a key role in identifying anomalies [34], providing early warnings to prevent collision incidents [35], mitigating navigational hazards [36], and bolstering maritime safety [37, 38]. STP research is bifurcated into short-term and long-term forecasts. The former focuses on immediate changes in a vessel's location and velocity to facilitate swift navigational adjustments and optimize maritime operations. These predictions are vital for real-time strategic decision-making, including maneuvering to avoid collisions, route optimization, and enhancing the overall safety and efficiency of maritime activities. Long-term forecasts extend beyond immediate positional and speed variations to include broader navigational trends and destinations, aiding in the planning of extended voyages, estimating arrival times, optimizing port visits, and more [39]. Thus, the development of long-term trajectory prediction models is increasingly prioritized, integrating motion patterns, intent forecasting, and contextual insights to improve prediction accuracy.

Recent advancements in trajectory prediction have seen a shift towards data-driven and deep learning approaches across various domains. [40] proposed a vehicle trajectory prediction framework using LSTM networks, demonstrating promising results in estimating future trajectories from highway driving data. Similarly, [41] developed a hybrid approach combining Principal Component Analysis (PCA) and GRU networks for IoT device mobility predictions in urban areas, claiming state-of-the-art results. [42] addressed pedestrian trajectory prediction in crowded spaces using an LSTM model that outperformed traditional methods on several public datasets.

A comparative study presented in [43] of expert-based and deep learning approaches for trajectory modeling in shared spaces with mixed traffic. They proposed two models: GSFM, an expert-based model combining Social Force Model and Game theory, and LSTM-DBSCAN, a deep learning model. Their evaluation using real-world mixed traffic data highlighted that while both approaches can generate realistic predictions, they differ in handling collisions and mimicking heterogeneous behavior. Collectively, these studies underscore the effectiveness of recurrent neural network architectures like LSTM and GRU in capturing complex behaviors and interactions in dynamic environments, spanning applications from autonomous vehicles to crowd management and IoT device tracking. A list of abbreviations used in this paper presented in Table 1.

## 3 Methodology

The methodology of our research is designed to enhance the precision of ship trajectory prediction through the innovative application of deep learning techniques. This process involves several step, initially we preprocess large raw AIS dataset. The AIS, which records ship trajectory data, is extremely large and is stored in CSV files. Each file contains data for thousands of ships over a 24-hour period, with records taken every seconds. Extracting data for a longer duration, such as three months, is quite challenging due to the sheer size and detail of the dataset. Managing this vast amount of data requires efficient techniques and tools to ensure that the extraction and processing are both accurate and timely. After analyzing the extensive dataset, individual ship data is extracted during the extraction phase, encompassing a duration of three months for each ship. This extracted data then undergoes preprocessing, where outliers

**Table 1. Abbreviations used in this paper.**

| Abbreviation | Definition |
|---|---|
| AIS | Automatic Identification System |
| CNN | Convolutional Neural Network |
| DNN | Deep Neural Network |
| LSTM | Long Short Term Memory |
| GRU | Gated Recurrent Neural Network |
| GPS | Global Positioning System |
| GIS | Geographic Information System |
| IMO | International Maritime Organization |
| MMSI | Maritime Mobile Service Identity |
| SOG | Speed Over Ground |
| COG | Course Over Ground |
| ADE | Average Displacement Error |
| FDE | Final Displacement Error |
| NL-ADE | Non-Linear Average Displacement Error |
| MSE | Mean Squared Error |
| MAE | Mean Absolute Error |

are removed, the data is down-sampled to one-hour intervals, and normalization is performed to standardize the dataset.

The clean data undergoes a training process where the model is trained using input features such as latitude (LAT), longitude (LON), speed, and heading. These features are used to teach the model how to predict the future trajectory. During prediction, based on the learned patterns from the training data, the model outputs the anticipated LAT and LON coordinates, which together define the desired trajectory of the ship, the whole process is presented in Fig 1. This process ensures that the model can accurately forecast the path or route based on the given input parameters., and finally we compare the model performance with other existing methods to validate our performance.

## 3.1 Data preprocessing

This research presents unique preprocessing strategies tailored specifically for AIS trajectory data [44]. Conventional trajectory preprocessing methods involve operations such as pinpointing stay points, reducing noise, and compressing trajectories [45]. However, to ensure the accuracy of AIS trajectories and the contextual significance within these trajectories, further refinement of AIS data is necessary [46].

The initial phase involves a comprehensive preprocessing of the raw AIS data. Given the common issues of missing values, ambiguous coordinates, and the presence of outliers in raw AIS data, our approach incorporates advanced preprocessing techniques. These techniques include data cleaning algorithms to rectify or remove inaccurate latitude and longitude information, interpolation methods to address missing values, and anomaly detection algorithms to identify and eliminate outliers. This step ensures that the data fed into the model is of the highest quality, thereby laying a solid foundation for accurate trajectory prediction. Fig 2 displays the real-time data gathered from the AIS device, as well as the trajectory extraction, AIS validity filtering, noise filtering, and the final preprocessed trajectory.

The dataset analyzed in Table 2 is a single ship data comprises 108,745 data points, providing a comprehensive view of vessel movements in a specific maritime region. The geographical

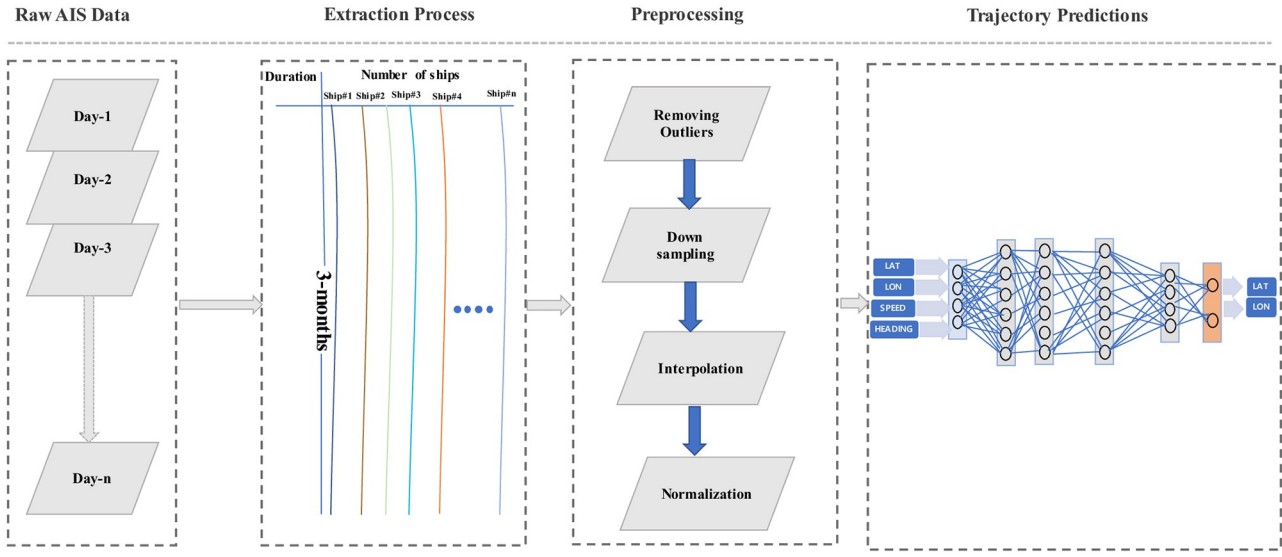

**Fig 1. Input to the deep learning model and output trajectory.**

scope spans LAT from 27.86˚N to 39.00˚N and LON from 121.10˚E to 131.80˚E, indicating a focus on East Asian waters. Vessel speeds range from stationary (0 knots) to 18.2 knots, with a mean speed of 14.54 knots and a standard deviation of 1.72 knots. The majority of vessels (50%) operate between 14.0 and 15.4 knots, suggesting common cruising speeds or operational

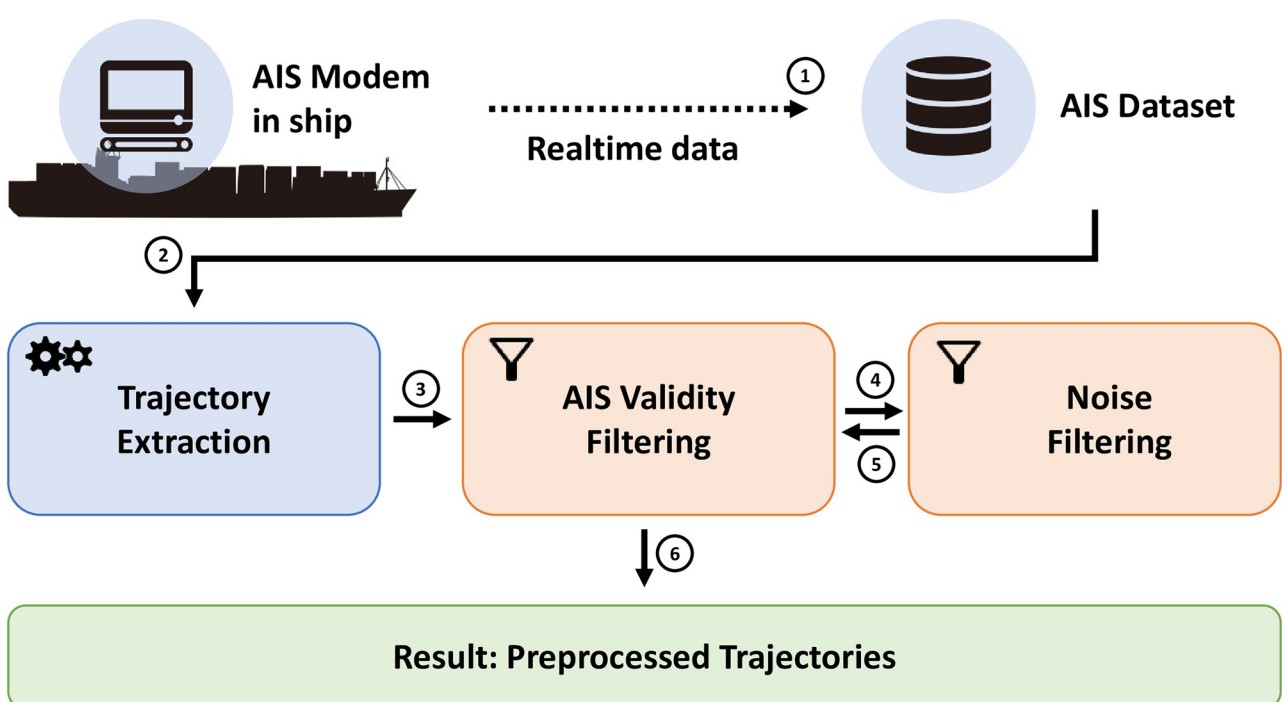

**Fig 2. Process of real-time data gathered from the AIS device, as well as the trajectory extraction, AIS validity filtering, noise filtering, and the final preprocessed trajectory.**

**Table 2. Statistical summary of AIS data.**

|       | LAT | LON | SPEED | HEADING |
|-------|-----|-----|-------|---------|
| count | 108745.000 | 108745.000 | 108745.000 | 108745.000 |
| mean  | 34.550 | 126.043 | 14.544 | 191.914 |
| std   | 1.505 | 2.006 | 1.723 | 91.308 |
| min   | 27.860 | 121.100 | 0.000 | 0.200 |
| 25%   | 34.000 | 124.900 | 14.000 | 100.400 |
| 50%   | 34.100 | 125.940 | 14.800 | 152.000 |
| 75%   | 35.160 | 127.300 | 15.400 | 272.500 |
| max   | 39.000 | 131.800 | 18.200 | 359.000 |

norms in the area. Heading data covers the full 360-degree range, reflecting diverse vessel trajectories. The interquartile range for LAT (34.00˚N to 35.16˚N) is relatively narrow, implying concentrated vessel activity within a specific latitudinal band, while the LON shows slightly more variation (IQR: 124.90˚E to 127.30˚E). This statistical overview provides insights into vessel movement patterns, potential high-traffic areas, and the intensity of maritime activity in the studied region, serving as a foundation for more detailed analyses of maritime traffic and behavior.

The preprocessed AIS data, visualized in Fig 3, reveals a clear and coherent vessel trajectory within the East Asian maritime region. The plot displays latitude ranging from 124.5˚E to 131˚E and longitude from 31˚N to 34.2˚N, aligning closely with the statistical summary presented in Table Y. The majority of data points form a distinct linear path from southwest to northeast, indicating the primary route taken by the tracked vessel(s). This linear trajectory demonstrates the effectiveness of the preprocessing steps in removing noise and erroneous data points while preserving the essential movement pattern. Notably, the visualization confirms that the preprocessed data falls within the expected ranges outlined in the statistical summary, with no points exceeding the minimum and maximum values previously reported. A small cluster of points around 34˚N latitude, appearing separate from the main trajectory, may represent stationary periods, different vessels, or minor outliers identified but not removed during preprocessing. The varying density of points along the main trajectory suggests fluctuations in vessel speed or data collection frequency. Overall, this visual representation of the preprocessed AIS data corroborates the statistical analysis and illustrates the successful application of data cleaning techniques, resulting in a dataset that accurately reflects vessel movements in the studied maritime area.

## 3.2 Knowledge extraction

Following the preprocessing, the cleaned AIS data undergoes a knowledge extraction phase. In this phase, we analyze the data to identify and extract ship trajectories. This involves segmenting the AIS data based on time intervals and geographical zones to construct meaningful trajectory patterns. Each segment represents a distinct path taken by ships, categorized according to specific areas of operation. This segmentation is critical for understanding the spatial-temporal patterns of ship movements, which is essential for accurate trajectory forecasting. Fig 4 shows the conceptual diagram of our proposed methodology.

**Algorithm 1**: Enhanced AIS Data Preprocessing for Vessel Trajectory Extraction

**Input**: $\mathcal{F} = \{f_1, f_2, \ldots, f_n\}$: Set of raw AIS data files
$\mathcal{M} = \{id_1, id_2, \ldots, id_m\}$: Set of vessel MMSI identifiers
**Output**: $T_{processed}$: Processed vessel trajectories file
1 **foreach** $id \in \mathcal{M}$ **do**
2 $T_{id} \leftarrow \emptyset$; // Initialize empty trajectory for vessel id

```
3   foreach f ∈ F do
4       D_f ← READCSV(f);
5       D'_f ← PREPROCESSDATA(D_f, id);
6       T_id ← T_id ∪ D'_f;
7   T_processed ← T_processed ∪ T_id;
8 EXPORTTOCSV(T_processed, 'ship_trajs.csv');
9 Function PREPROCESSDATA(D, id):
10  D ← {(lon, lat, speed, heading) ∈ D | 120° ≤ lon ≤ 140° ∧ 20° ≤
    lat ≤ 40°};
11  D ← {(lon, lat, speed, heading) ∈ D | speed ≤ 40 knots ∧
    heading ≤ 360°};
12  D ← {(lon, lat, speed, heading) ∈ D | vessel_id = id};
13  D ← RESAMOKEHOURLY(D);
14  D ← REMOVEDUPLICATES(D);
15  D ← INTERPOLATEGAPS(D);
16  return D;
```

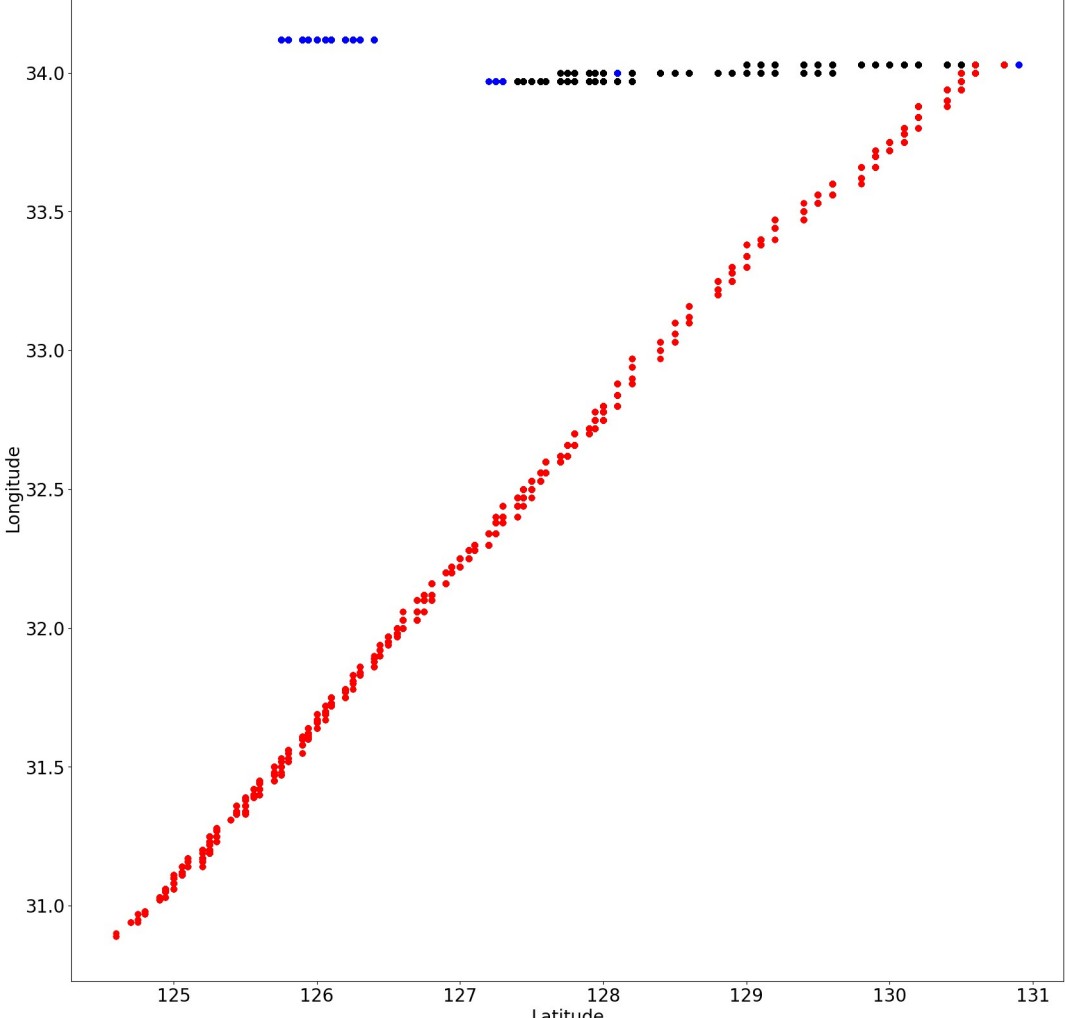

**Fig 3. Preprocessed AIS trajectory data plotted as latitude versus longitude.** This visualization demonstrates the effectiveness of data preprocessing in producing a coherent representation of maritime movement patterns in the East Asian region.

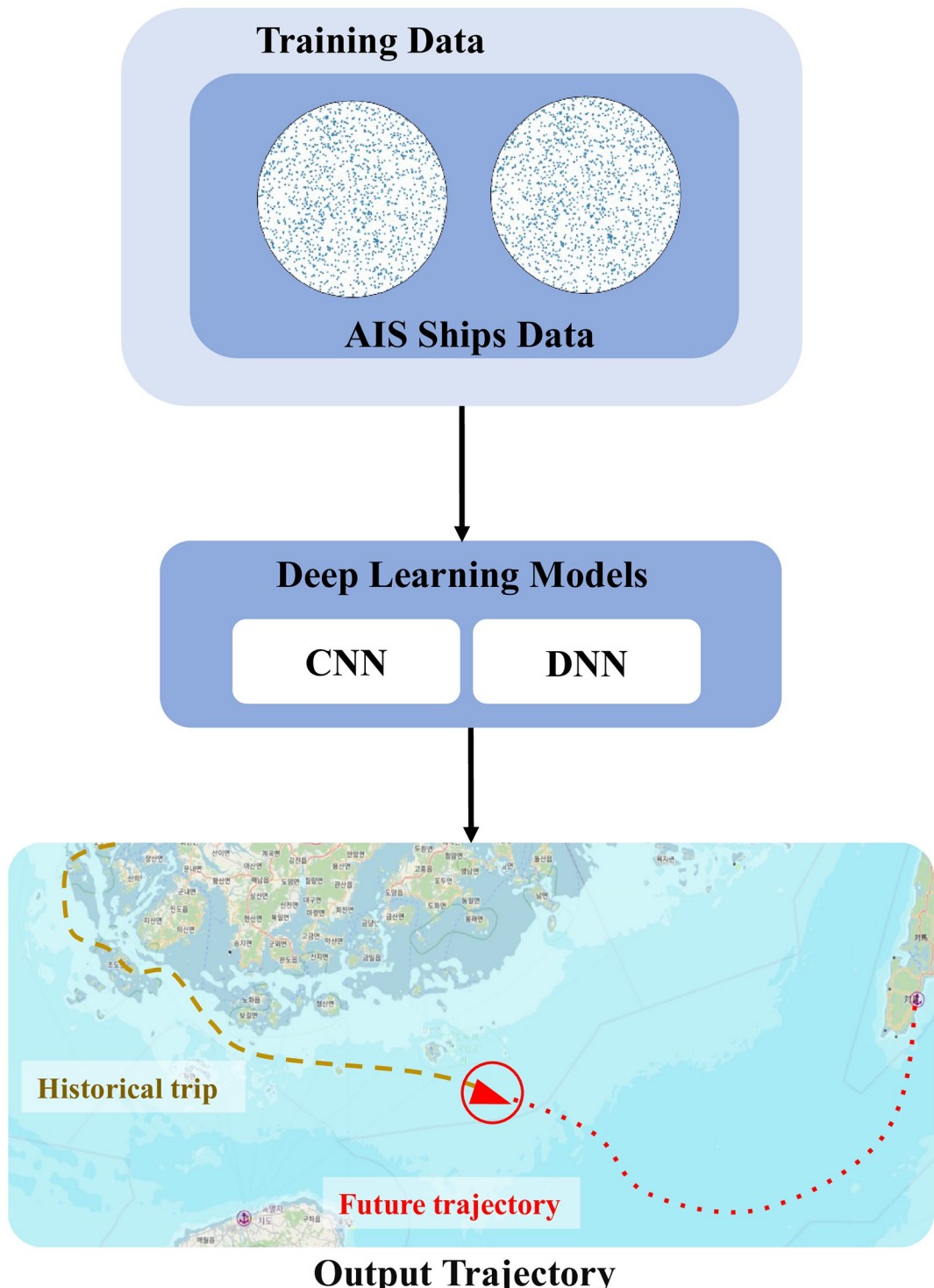

**Fig 4. Conceptual structure diagram of the proposed model.**

The proposed data preprocessing algorithm efficiently extracts and refines vessel trajectories from raw AIS data. It iterates through a set of vessel identifiers (MMSI numbers) and processes multiple AIS data files for each vessel. The algorithm employs a dedicated preprocessing function that filters data based on geographical boundaries (LON 120˚-140˚, LAT 20˚-40˚), removes anomalous speed and heading values, and focuses on the specific vessel being processed. It then applies temporal resampling to achieve hourly intervals, eliminates duplicate entries, and interpolates gaps in the trajectory data. This approach ensures the resulting trajectories are clean, consistent, and suitable for further analysis. By processing each vessel's data across all input files before moving to the next vessel, the algorithm maintains efficiency while producing a comprehensive dataset of processed trajectories. The final output is a consolidated CSV file containing refined trajectory data for all vessels, ready for subsequent analytical tasks or modeling efforts in maritime traffic analysis.

## 3.3 Trajectory prediction modeling

In our proposed methodology, we have utilized advanced deep learning techniques, namely CNN and DNN. These models are thoroughly structured with several layers to adeptly manage the complexities involved in predicting ship trajectories. By feeding segmented trajectory data into our models, the model learns from past trajectory history and forecast the future positions of ships.

**3.3.1 CNN model for vessel trajectroy prediction.**   In our proposed methodology, we introduce an advanced 1D-CNN architecture specifically designed for vessel trajectory prediction, a challenging domain where traditional models often fail to accurately capture the dynamic patterns and shifts within maritime navigation data. This architecture is meticulously designed with 32 convolution filters, each with a kernel size of 3, which plays a vital role in analyzing and understanding the complex, multidimensional structures inherent in datasets comprising latitude, longitude, speed, and heading parameters. Each observation at a given time $t$ is represented as:

$$x_t = [\text{lat}_t, \text{lon}_t, \text{speed}_t, \text{heading}_t] \tag{1}$$

and a sequence of such observations over $T$ time steps forms a trajectory:

$$\text{Traj} = [x_1, x_2, \ldots, x_T] \tag{2}$$

The input to our model consists of 3 hours of historical data, downsampled to 1-hour intervals, creating an input sequence $\mathbf{X}$:

$$\mathbf{X} = [x_{t-2}, x_{t-1}, x_t] \tag{3}$$

where $\mathbf{X} \in \mathbb{R}^{3 \times 4}$ includes 3 time steps and 4 features per time step. The 1D-CNN employs a convolutional kernel of size 3, denoted as $\mathbf{W}$:

$$\mathbf{W} \in \mathbb{R}^{3 \times 4 \times 32} \tag{4}$$

where 32 represents the number of filters. The convolution operation, initiated with a 'sigmoid' activation function, is mathematically expressed as:

$$\mathbf{Y} = \sigma(\mathbf{W} * \mathbf{X} + \mathbf{b}) \tag{5}$$

where * denotes convolution, $\sigma$ is the sigmoid activation function, and $\mathbf{Y} \in \mathbb{R}^{1 \times 32}$ is the resultant feature map. This configuration enables the model to perform non-linear transformations and extract a wide range of features from the complex input data related to vessel movements. To predict the next time step's latitude and longitude, the output feature map $\mathbf{Y}$ is passed

through a fully connected (dense) layer:

$$[\text{lat}_{t+1}, \text{lon}_{t+1}] = \mathbf{W}_{\text{fc}} \cdot \mathbf{Y} + \mathbf{b}_{\text{fc}} \qquad (6)$$

where $\mathbf{W}_{\text{fc}} \in \mathbb{R}^{32 \times 2}$ and $\mathbf{b}_{\text{fc}} \in \mathbb{R}^{2}$. This formulation enables the model to effectively capture temporal dependencies and predict future positions, leveraging the convolutional layers for feature extraction and the dense layer for final position prediction. The use of 1-hour interval data with a history of 3 hours ensures that the model has sufficient context to make accurate predictions for the next hour.

To address the persistent challenge of overfitting—a common obstacle in deep learning models within the context of maritime trajectory analysis—our architecture incorporates a strategic blend of max pooling and dropout techniques. Max pooling helps to streamline the model by reducing the spatial dimensions of the representation, thereby decreasing the computational load, and focusing on the most significant navigational features. Simultaneously, dropout serves as a form of regularization, randomly omitting a subset of the features in each iteration. This makes the model learn more robust and generalized patterns, preventing it from over-relying on a limited set of heavily weighted inputs. Further augmenting the model's efficiency and effectiveness, we employ batch normalization, a method that standardizes the inputs to a layer for each mini batch. This technique ensures the stability of the learning process, speeds up the training phase, and frequently results in notable performance enhancements by minimizing internal covariate shift. This normalization step guarantees that our model remains flexible and responsive throughout the training process, adept at handling the varied and dynamic nature of maritime trajectory data.

The architecture comprises of two fully connected layers, which are involved in converting the learned features into actionable understandings for predicting vessel trajectories. These layers utilize the SoftMax function to compute the probability distribution over potential future positions, offering a precise, probabilistic framework for forecasting. This approach enables our model to not only predict the most probable future trajectory but also to assess the confidence in its predictions, a critical advantage in the practical application of vessel trajectory prediction. By integrating these complex techniques and architectural decisions, our proposed 1D-CNN model emerges as a comprehensive and powerful solution for predicting vessel trajectories. Fig 5 shows the complete structure diagram of our proposed 1D-CNN model with historical data and target data.

### 3.4 Comparative analysis and model evaluation

Comparative analysis of the CNN and DNN models is an integral part of this research. By evaluating these models against various performance metrics, such as MAE, MSE, ADE, FDE and

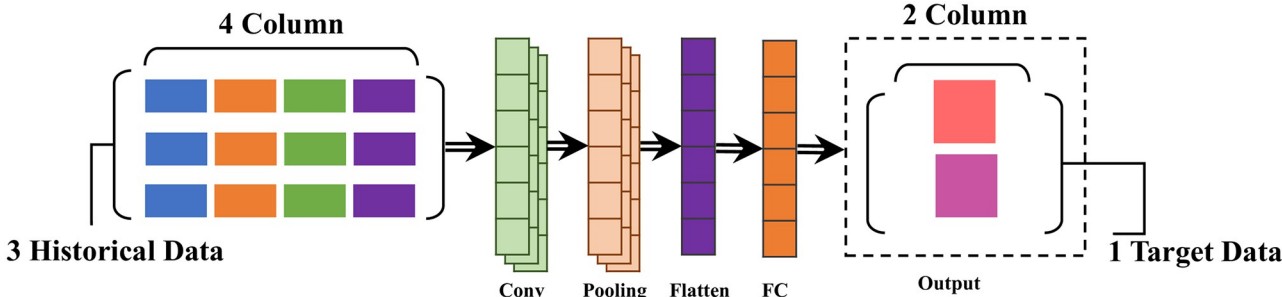

**Fig 5. Complete structure diagram of our proposed 1D-CNN model with historical data and target data.**

NL-ADE we aim to determine their effectiveness in trajectory prediction. This evaluation not only involves assessing the accuracy of predictions but also explores the models' ability to generalize across different regions and conditions. The ultimate goal is to identify the model that provides the most reliable and accurate trajectory forecasts, thereby contributing valuable insights to the field of maritime navigation and predictive modeling. Mathematically, it is expressed as:

$$\text{MAE} = \frac{1}{n} \sum_{i=1}^{n} |x_i - \hat{x}_i| \tag{7}$$

where $n$ is the number of observations, $x_i$ is the actual value for the $i^{th}$ observation, $\hat{x}_i$ is the predicted value for the $i^{th}$ observation, $|\cdot|$ denotes the absolute value.

MSE gives more weight to larger errors. It is defined as:

$$\text{MSE} = \frac{1}{n} \sum_{i=1}^{n} (x_i - \hat{x}_i)^2 \tag{8}$$

where $n$ is the number of observations, $x_i$ is the actual value for the $i^{th}$ observation, $\hat{x}_i$ is the predicted value for the $i^{th}$ observation.

Both MAE and MSE are non-negative, where lower values indicate better model performance. MSE is more sensitive to outliers than MAE because it squares the errors before averaging them, which disproportionately increases the impact of large errors on the model's overall error metric.

## 4 Results and discussions

Experiments were conducted using Python and TensorFlow 2 on a high-performance computing machine running the Windows operating system. We utilized seven datasets, each corresponding to a different ship. The data from each ship spanned over three months and was divided into training, validation, and testing sets with proportions of 70%, 15%, and 15%, respectively.

The performance evaluation of the CNN and DNN models on the AIS datasets involved rigorous training and testing phases. For the training phase, we utilized a variety of hyperparameters to optimize the models. The final values for each hyperparameter, at which both models converged, are as follows: epochs set to 900, learning rate value set to 0.0005. For the CNN model, the number of filters is set to 128, and another Dense layer with 32 neurons is utilized. In the case of the DNN model, four layers with 64, 32, 16, and 2 neurons, respectively, are employed, with ReLU activation function. This optimization process was critical, as it significantly influenced the accuracy and efficiency of the models in predicting ship trajectories. The CNN model, known for its ability to capture spatial hierarchies in data, was particularly adept at recognizing patterns in the spatial distribution of AIS data points. This capability made it highly effective in understanding complex maritime traffic scenarios and predicting the future locations of ships with high precision.

On the other hand, the DNN model, with its deep architecture, excelled in capturing the temporal dependencies of the AIS data. By processing sequential data points, the DNN was able to infer the dynamic behavior of ships, including changes in speed and direction over time. This understanding enabled the DNN to forecast ship trajectories with notable accuracy, although it sometimes struggled with highly erratic movements that are characteristic of certain maritime activities.

**Table 3. Model performance for different ships.**

| Ship ID | Model | Training Loss | | Validation Loss | | Additional Metrics | | |
|---------|-------|------|------|------|------|------|------|------|
| | | MSE | MAE | MSE | MAE | ADE | FDE | NL-ADE |
| A0001 | CNN | 0.035 | 0.102 | 0.023 | 0.084 | 0.349 | 0.458 | 0.388 |
| | DNN | 0.035 | 0.124 | 0.089 | 0.222 | 0.621 | 1.567 | 0.622 |
| A0002 | CNN | 0.035 | 0.148 | 0.152 | 0.274 | 0.477 | 0.473 | 0.479 |
| | DNN | 0.051 | 0.186 | 0.087 | 0.230 | 0.643 | 0.598 | 0.650 |
| A0003 | CNN | 0.016 | 0.087 | 0.007 | 0.063 | 0.140 | 0.042 | 0.143 |
| | DNN | 0.049 | 0.177 | 0.070 | 0.229 | 0.546 | 0.516 | 0.546 |
| A0004 | CNN | 0.005 | 0.063 | 0.005 | 0.057 | 0.176 | 0.068 | 0.187 |
| | DNN | 0.027 | 0.138 | 0.029 | 0.149 | 0.512 | 0.491 | 0.516 |
| A0005 | CNN | 0.017 | 0.075 | 0.016 | 0.075 | 0.259 | 0.152 | 0.263 |
| | DNN | 0.046 | 0.167 | 0.048 | 0.171 | 0.676 | 0.302 | 0.674 |
| A0006 | CNN | 0.011 | 0.083 | 0.021 | 0.105 | 0.345 | 0.310 | 0.349 |
| | DNN | 0.018 | 0.106 | 0.011 | 0.083 | 0.672 | 0.577 | 0.679 |
| A0007 | CNN | 0.057 | 0.120 | 0.034 | 0.100 | 0.183 | 0.134 | 0.183 |
| | DNN | 0.158 | 0.183 | 0.018 | 0.086 | 0.553 | 0.583 | 0.554 |

To quantify the performance of both models, CNN and DNN across all ships reveals notable distinctions in their efficacy Table 3. Overall, the CNN models tend to exhibit lower mean squared error (MSE) and mean absolute error (MAE) values on both training and validation datasets compared to their DNN counterparts, indicating superior predictive accuracy. For instance, across all ships, CNN models consistently achieve lower MSE and MAE values on both training and validation datasets, with notable examples including Ship A0001 (CNN: MSE = 0.0349, MAE = 0.1016; DNN: MSE = 0.0350, MAE = 0.1235) and Ship A0004 (CNN: MSE = 0.0053, MAE = 0.0634; DNN: MSE = 0.0266, MAE = 0.1384). This trend suggests that CNN architectures are generally better suited for capturing complex patterns and relationships in the data, leading to more accurate predictions across different ship contexts. The results indicated that while both models performed commendably, the CNN model generally outperformed the DNN model in terms of spatial predictions. This was attributed to its superior ability to process and interpret the spatial features of the AIS data.

Conversely, for temporal predictions involving the estimation of a ship's future positions based on its past trajectory, the DNN model demonstrated a slight advantage over the CNN model. This was largely due to its deeper architecture, which was more adept at learning and remembering long-term dependencies in the data. Despite these differences, both models showed a significant improvement over traditional trajectory prediction methods, highlighting the potential of deep learning techniques in enhancing maritime surveillance and navigation systems. Furthermore, we explored the impact of data augmentation techniques, such as rotation and scaling of the AIS data points, on model performance. These techniques were aimed at increasing the robustness of the models by exposing them to a wider variety of training scenarios. The findings suggested that data augmentation contributed to a noticeable improvement in the models' ability to generalize, thereby reducing overfitting and enhancing prediction accuracy on the test set.

Fig 6 illustrates the actual and predicted trajectory output results for analysis. Each pair of figures, marked from (a) through (n), provides a visual representation of the trajectories predicted by two distinct models. Specifically, the trajectory on the left side of each pair corresponds to the output of the CNN-based model, while the trajectory on the right side represents the result of the DNN-based model. This arrangement allows for a direct comparison between

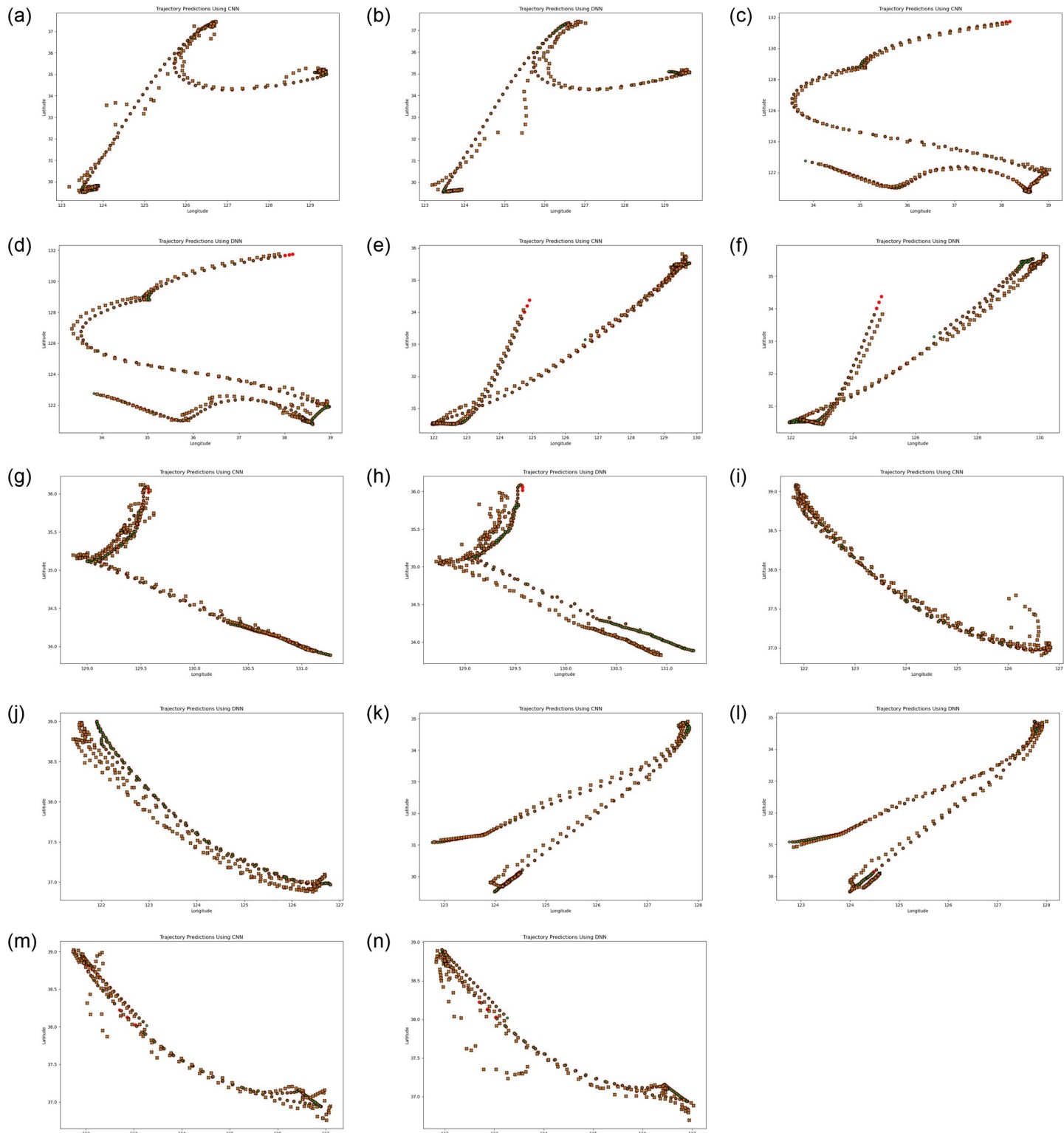

**Fig 6.** (a) to (n) shows the Actual trajectories and predicted trajectories for different ship IDs using CNN and DNN. (a) Ship ID: A0001—CNN, (b) Ship ID: A0001—DNN, (c) Ship ID: A0002—CNN, (d) Ship ID: A0002—DNN, (e) Ship ID: A0003—CNN, (f) Ship ID: A0003—DNN, (g) Ship ID: A0004—CNN, (h) Ship ID: A0004—DNN, (i) Ship ID: A0005—CNN, (j) Ship ID: A0005—DNN, (k) Ship ID: A0006—CNN, (l) Ship ID: A0006—DNN, (m) Ship ID: A0007—CNN, (n) Ship ID: A0007—DNN.

the predictive capabilities of the two models. With a total of 14 figures, evenly split between CNN and DNN trajectories, the visualizations serve as a comprehensive examination of the models' performance in trajectory prediction.

In assessing the models' performance, a range of metrics was employed, prominently including MSE and MAE. These metrics were pivotal in gauging the accuracy and reliability of the trajectory predictions generated by the models. Computation of these metrics was conducted both during the training phase and the subsequent validation phase, providing insights into the models' capacity to generalize to new data. The outcomes of this evaluation revealed discernible disparities in the performance of CNN and DNN models. Notably, CNN models consistently exhibited lower training and validation losses when compared to their DNN counterparts. Such results underscored the superior predictive accuracy inherent in CNN models, emphasizing their efficacy in handling the intricate spatial complexities intrinsic to trajectory prediction tasks.

The Table 3 includes ADE, FDE, and NL-ADE, providing a thorough evaluation of predictive accuracy for CNN) and DNN across different ships. ADE assesses the average distance between predicted and actual trajectories, while FDE measures deviation at the final time step. NL-ADE normalizes ADE for scale, enabling standardized comparison.

Analysis shows that CNN generally outperforms DNN in both ADE and FDE, indicating better overall accuracy and final prediction precision. Although NL-ADE values are similar for both models, suggesting comparable performance when normalized, CNNs consistently deliver lower ADE and FDE values. Hence, CNNs are preferred for precise trajectory forecasting due to their superior performance in minimizing prediction errors.

Figs 7 and 8 further illustrates the comparison between CNN and DNN models in terms of MSE and MAE, highlighting the CNN model's superior performance. The lower MSE and MAE values for the CNN model suggest that it was better at capturing the nuances of the trajectory data, leading to more accurate predictions. This can be attributed to CNN's ability to

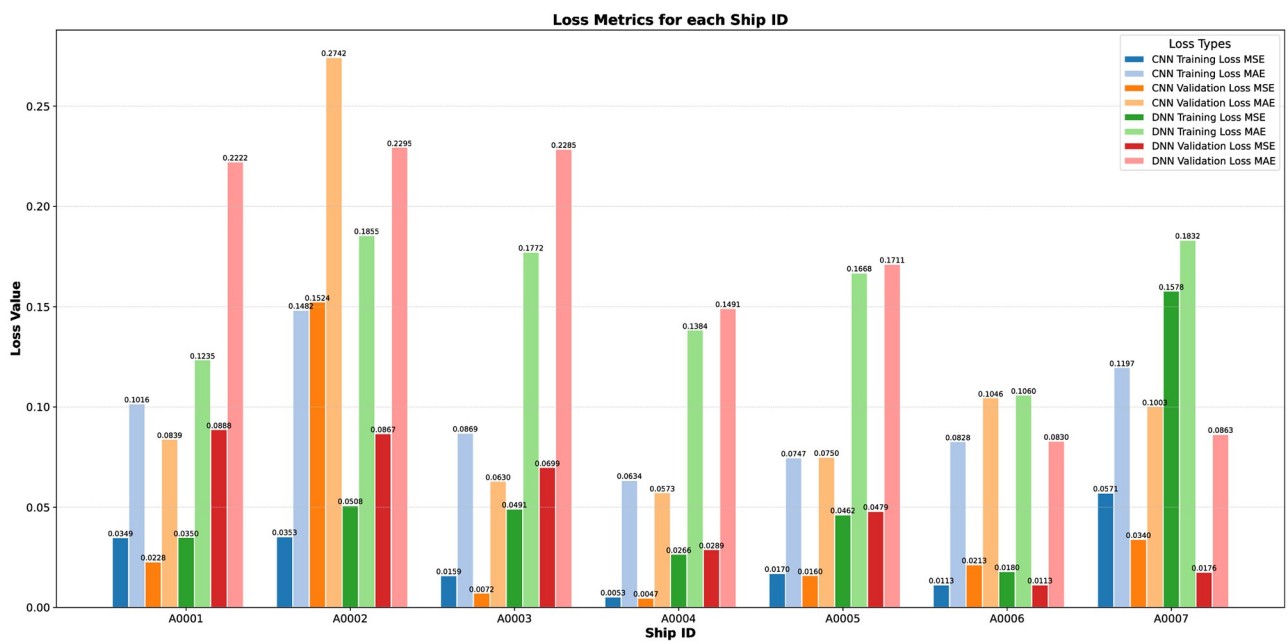

**Fig 7. Comparison of actual trajectories and predicted trajectories for different ship IDs using CNN and DNN.**

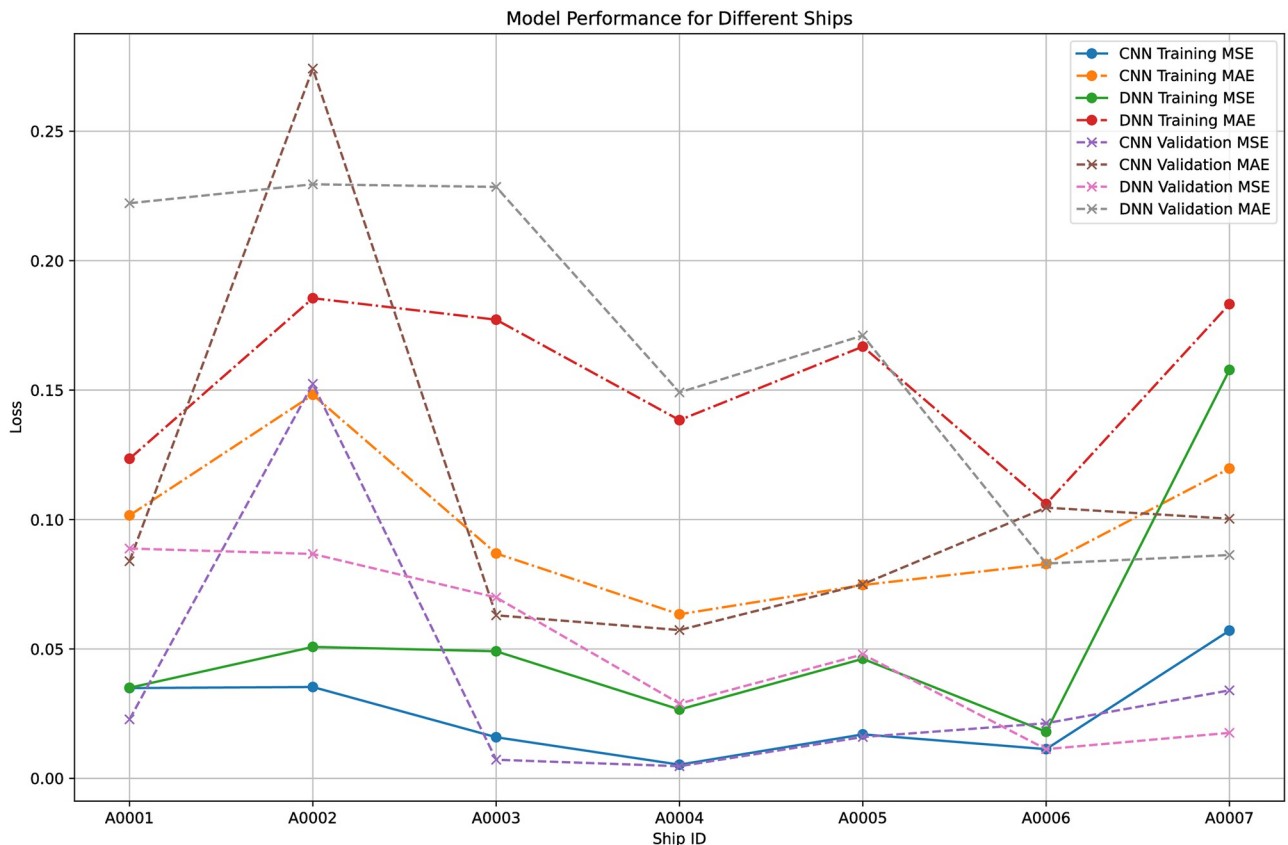

**Fig 8. Perfromance comparison of each model with different ship ID.**

effectively process spatial information and recognize patterns within the data, which is crucial for accurately forecasting the movements of objects, such as ships in this context.

Additionally, the analysis delved into the models' behavior over different phases of the training process, examining how quickly and effectively each model learned from the data. The CNN model not only demonstrated a faster convergence towards lower error rates but also maintained a steady improvement, suggesting a robust learning capability. In contrast, the DNN model, while still showing significant learning, exhibited a slower pace of improvement and occasionally struggled with complex spatial relationships within the data.

The superiority of the CNN model in this experimental setup can be linked to its architectural advantages for processing and learning from image-like data structures, which are common in trajectory prediction tasks. The convolutional layers in CNNs are particularly effective at extracting and learning hierarchical feature representations from such data, making them more suited for tasks that involve spatial reasoning and prediction. However, it's important to note that the DNN model also showed considerable predictive capabilities, particularly in scenarios where temporal dynamics were more pronounced than spatial variations. This suggests that while CNNs may be the preferred model for spatially complex prediction tasks, DNNs could be more suitable for scenarios where understanding temporal patterns and dependencies is crucial. Table 4 shows the training and validation losses for ship IDs A-00008 and A-00009 using LSTM and GRU models.

**Table 4. Training and validation losses for ship IDs ABX-00004 and ABX-00005 using LSTM and GRU models.**

| Ship ID | Model | MSE (Train) | MAE (Train) | MSE (Val) | MAE (Val) |
|---------|-------|-------------|-------------|-----------|-----------|
| A-00008 | LSTM | 0.025 | 0.128 | 0.025 | 0.128 |
|         | GRU  | 0.013 | 0.085 | 0.012 | 0.082 |
| A-00009 | LSTM | 0.015 | 0.091 | 0.018 | 0.098 |
|         | GRU  | 0.005 | 0.011 | 0.002 | 0.013 |

## 4.1 Comparison with other existing methods

The comparison between the proposed method and existing models are shown in Table 5, including RNN [40], GRU [41], LSTM [42], and DBS-LSTM [43], highlights significant improvements in prediction accuracy and handling of non-linear trajectories. The metrics used for evaluation—ADE, FDE, and NL-ADE—are crucial in assessing the performance of trajectory prediction models.

The RNN model, with an ADE of 0.587 ± 0.132 and an FDE of 0.639 ± 0.155, demonstrates lower accuracy compared to more advanced methods. Its NL-ADE of 0.501 ± 0.108 also indicates limitations in managing non-linear trajectory components. The GRU model shows improvement, with a reduced ADE of 0.491 ± 0.090 and an FDE of 0.557 ± 0.127, along with a better NL-ADE of 0.418 ± 0.091, suggesting enhanced capability in handling non-linearities.

The LSTM model further enhances prediction accuracy, achieving an ADE of 0.428 ± 0.108 and an FDE of 0.472 ± 0.114. Its NL-ADE of 0.376 ± 0.097 reflects a stronger performance in managing complex trajectory patterns. The DBS-LSTM model stands out with the lowest ADE (0.307 ± 0.097) and FDE (0.328 ± 0.082) among the compared methods, and its NL-ADE of 0.260 ± 0.112 highlights its robustness in handling non-linear segments. The proposed method, with an ADE of 0.35998 ± 0.15977, demonstrates a balanced performance, significantly improving upon traditional RNN and GRU models. Its FDE of 0.28449 ± 0.20699 indicates superior accuracy in predicting final positions, which is crucial for applications requiring precise endpoint predictions. The NL-ADE of 0.37013 ± 0.15945 shows the model's adeptness at handling non-linear trajectory components, making it competitive with the DBS-LSTM model.

Overall, the proposed method offers a promising alternative to existing models, particularly excelling in FDE as presented in Fig 9 and demonstrating robust capabilities in managing non-linear trajectories. This balanced performance across all metrics underscores the method's potential for applications where accurate final position prediction and handling of non-linear aspects are critical.

## 5 Conclusions

In conclusion, this research has presented a novel and effective ship trajectory prediction method leveraging CNN, DNN, LSTM, and GRU. Through accurate preprocessing of a

**Table 5. Comparison of metrics across different methods.**

| Methods | ADE | FDE | NL-ADE |
|---------|-----|-----|--------|
| RNN [40] | 0.587 ± 0.132 | 0.639 ± 0.155 | 0.501 ± 0.108 |
| GRU [41] | 0.491 ± 0.090 | 0.557 ± 0.127 | 0.418 ± 0.091 |
| LSTM [42] | 0.428 ± 0.108 | 0.472 ± 0.114 | 0.376 ± 0.097 |
| DBS-LSTM [43] | 0.307 ± 0.097 | 0.328 ± 0.082 | 0.260 ± 0.112 |
| Proposed Method | 0.35998 ± 0.15977 | 0.28449 ± 0.20699 | 0.37013 ± 0.15945 |

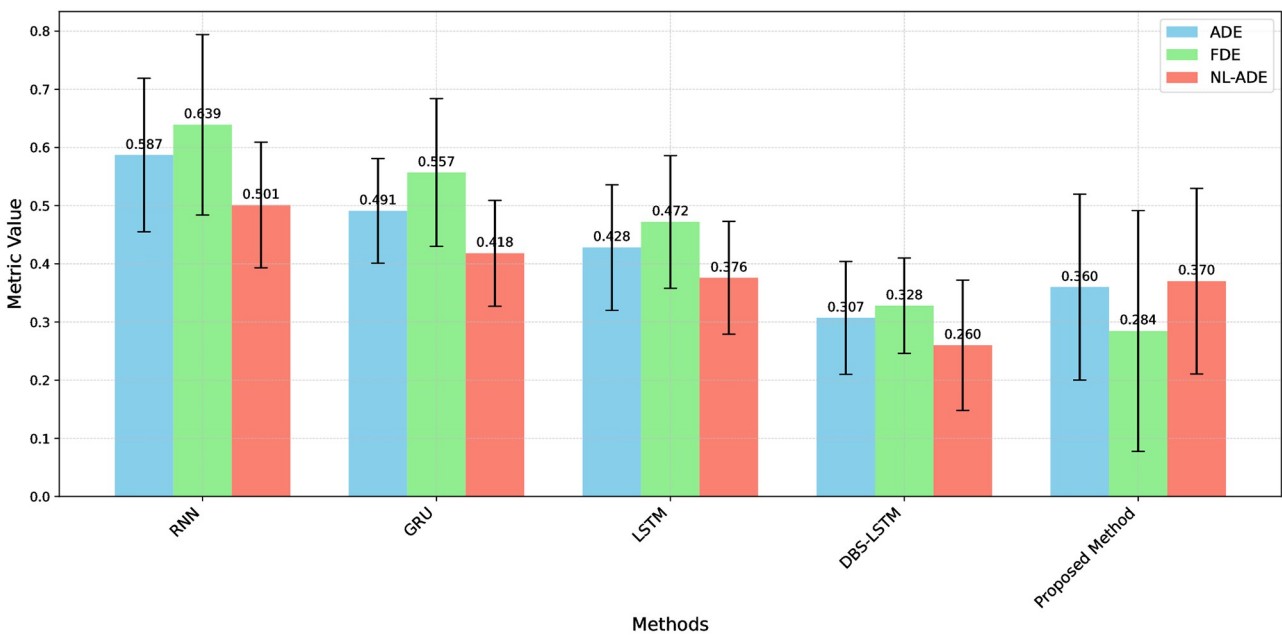

**Fig 9. Perfromance comparison of each model.**

substantial AIS dataset, valuable features were extracted to inform the predictive capabilities of these models. The application of CNN and DNN demonstrated notable success in capturing and learning the inherent patterns within ship trajectory data, resulting in a substantial reduction in MAE and MSE compared to other deep learning models. Experimental results showed that the CNN model significantly reduced both MAE and MSE, showcasing superior performance overall when compared to other deep learning algorithms.

Additionally, a comparative analysis with other models—RNN, GRU, LSTM, and DBS-LSTM—using metrics such as ADE, FDE, and NL-ADE demonstrates the robustness and accuracy of our proposed method. This advancement in accuracy holds significant implications for the enhancement of ship trajectory prediction, thereby contributing to the promotion of maritime traffic safety.

Future research could explore hybrid approaches integrating CNN and other deep learning methodologies to further enhance prediction accuracy and robustness. These findings underscore the efficacy of CNN models in analyzing and predicting ship trajectories while also highlighting the potential for improvement within the realm of maritime navigation systems.

## Author Contributions

**Conceptualization:** Umar Zaman, Junaid Khan, Kyungsup Kim.

**Data curation:** Kyungsup Kim.

**Formal analysis:** Umar Zaman, Junaid Khan.

**Investigation:** Umar Zaman, Junaid Khan, Kyungsup Kim.

**Methodology:** Umar Zaman, Junaid Khan, R. Y. Aburasain.

**Project administration:** Kyungsup Kim.

**Resources:** Kyungsup Kim.

**Software:** Umar Zaman, Junaid Khan, Awatef Salim Balobaid.

**Supervision:** Kyungsup Kim.

**Validation:** Umar Zaman, Junaid Khan, Eunkyu Lee.

**Visualization:** Umar Zaman, Junaid Khan.

**Writing – original draft:** Umar Zaman, Junaid Khan.

**Writing – review & editing:** Eunkyu Lee, Awatef Salim Balobaid, R. Y. Aburasain.

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
