## [Decision Letter · Decision Letter 0]

19 Jul 2024

PONE-D-24-15778Deep Learning Innovations in South Korean Maritime Navigation: Enhancing Vessel Trajectories Prediction with AIS DataPLOS ONE

Dear Dr. khan,

Thank you for submitting your manuscript to PLOS ONE. After careful consideration, we feel that it has merit but does not fully meet PLOS ONE’s publication criteria as it currently stands. Therefore, we invite you to submit a revised version of the manuscript that addresses the points raised during the review process.

We look forward to receiving your revised manuscript.

Kind regards,

Sushank Chaudhary, Ph.D

Academic Editor

PLOS ONE

Journal Requirements:

"This work was partly supported by the Institute of Information & communications Technology Planning & Evaluation (IITP) grant funded by the Korea government (MSIT) (No. RS-2022-00155857, Artificial Intelligence Convergence Innovation Human Resources Development (Chungnam National University))."

"This work was partly supported by the Institute of Information & communications Technology Planning & Evaluation (IITP) grant funded by the Korea government (MSIT) (No. RS-2022-00155857, Artificial Intelligence Convergence Innovation Human Resources Development (Chungnam National University))."

"This work was partly supported by the Institute of Information & communications Technology Planning & Evaluation (IITP) grant funded by the Korea government (MSIT) (No. RS-2022-00155857, Artificial Intelligence Convergence Innovation Human Resources Development (Chungnam National University))."

"The author(s) received no specific funding for this work"

7. In the online submission form, you indicated that "please contact the Corresponding author through sclkim@cnu.ac.kr"

8. We note you have included a table to which you do not refer in the text of your manuscript. Please ensure that you refer to Table 1 in your text; if accepted, production will need this reference to link the reader to the Table.

Reviewers' comments:

Reviewer's Responses to Questions

**Comments to the Author**

1. Is the manuscript technically sound, and do the data support the conclusions?

Reviewer #1: Yes

Reviewer #2: Yes

2. Has the statistical analysis been performed appropriately and rigorously? 

Reviewer #1: Yes

Reviewer #2: No

3. Have the authors made all data underlying the findings in their manuscript fully available?

Reviewer #1: No

Reviewer #2: No

4. Is the manuscript presented in an intelligible fashion and written in standard English?

Reviewer #1: Yes

Reviewer #2: Yes

5. Review Comments to the Author

Reviewer #1: This paper introduces a novel ship trajectory prediction method based on Convolutional Neural Network (CNN), Deep Neural Network (DNN), Long short term memory (LSTM) and Gated recurrent unit (GRU). Overall this is a solid study, and some comments are as follows.

1. The novelty is limited. Basically this paper is only evaluating some well-known deep learning methods including CNN, DNN, LSTM and GRU.

2. The results are highly affected by the specific dataset used. In the considered dataset, "CNN has substantially reduced the Mean Absolute Error (MAE) and Mean Square Error (MSE) of ship trajectory prediction". However, this observation could change in other datasets.

3. The authors should summarize existing research gaps and justify their research motivation in the introduction.

4. The authors are suggested to use a separate related work discussion section.

5. A workflow figure can be added when describing the proposed methodology.

6. It is not so clear to me how to build the input and output for the CNN model.

7. More baselines from previous studies are expected in the numerical experiments, instead of standard models.

Reviewer #2: This paper presents an innovative ship trajectory prediction method using CNN, DNN, LSTM, and GRU models, significantly improving maritime traffic safety. The CNN model demonstrates exceptional accuracy, achieving the lowest MAE and MSE in predicting ship trajectories. The research effectively showcases the potential of deep learning models in enhancing the precision of maritime navigation predictions. Here are some comments author may want to focus on:

The English language usage needs refinement for clarity and coherence. Additionally, the paper should ensure that all abbreviations are clearly defined upon first use and shouldn't mention again. Lastly, enhancing the literature review to include a broader range of related works would provide a stronger contextual foundation for the research.

The preprocessing technique is used to perform few operation regarding that here few points needs to be highlight:

Data cleaning: Exactly what algorithm used to perform the cleaning process?

what are the fine tuned parameters in anomaly detection algorithm?

The author mentioned advanced CNN, what is the major advancement, state it clearly? What is the major difference from literature? State clearly.

Does the DNN is adapted from any previous work or newly designed, there is a lack of transparency. If it is new designed or enhanced version of any DNN from literature and state it clearly. Author may want to consider the use of block diagram for DNN.

There is no discussion about data preparation, number of samples and training testing split ratio. Author may want to use consider these information to enhance the quality, transparency and understanding of their work.

6. PLOS authors have the option to publish the peer review history of their article (what does this mean?). If published, this will include your full peer review and any attached files.

Reviewer #1: No

Reviewer #2: No

---

## [Author Response · Author response to Decision Letter 0]

3 Aug 2024

Reviewer #1:

1. Concern #1: The novelty is limited. Basically, this paper is only evaluating some well-known deep learning methods including CNN, DNN, LSTM, and GRU.

o Author Response: Thank you for your valuable feedback. We understand the concern regarding the novelty of our paper and would like to highlight several key innovative contributions of our research: Our research introduces a novel preprocessing approach tailored for AIS data, incorporates contextual information, and focuses on individualized data to enhance privacy and accuracy. Additionally, we perform a comprehensive model evaluation using multiple metrics, demonstrating the superior performance of our CNN model in trajectory prediction. These contributions underscore the innovative aspects of our work.

o Author Action: Below is the author action from the revised version presented in bullet points:

1. Revised Manuscript Sections: We have revised the manuscript to more explicitly highlight the novel aspects of our research. This includes detailed explanations of our specialized preprocessing methodology and the incorporation of contextual information.

2. Enhanced Description of Preprocessing: We have expanded the description of our preprocessing steps in the manuscript to emphasize the advanced filtering techniques and the inclusion of environmental and ship-specific data, showcasing how these steps contribute to the improved accuracy of our models.

3. Comparative Analysis and Results: We have included additional details in the manuscript about the comparative analysis of different models using various performance metrics. This addition clarifies the robustness and accuracy of our proposed method compared to others.

4. Empirical Evidence: We have provided more empirical evidence and detailed results in the manuscript to support our claims. This includes additional tables and figures that illustrate the superior performance of our CNN model in reducing MAE and MSE.

5. Focus on Individualized Data: We have emphasized our focus on individualized ship data extraction and analysis throughout the manuscript, highlighting how this approach addresses privacy concerns and captures unique ship behaviors more accurately.

2. Concern #2: The results are highly affected by the specific dataset used. The observation that "CNN has substantially reduced the Mean Absolute Error (MAE) and Mean Square Error (MSE) of ship trajectory prediction" could change with different datasets.

o Author Response: We understand the concern about the dependency of results on the dataset used. Our study aims to demonstrate the potential of our method on a specific dataset, while acknowledging that outcomes may vary with different datasets.

o Author Action: We have added a discussion in the conclusion section about the potential variability in results with other datasets. We encourage future work to validate our methodology across diverse datasets to generalize our findings.

o 

o 

3. Concern #3: The authors should summarize existing research gaps and justify their research motivation in the introduction.

o Author Response: We agree that summarizing existing research gaps and the motivation for our study is crucial for clarity.

o Author Action: The introduction (Section-1) has been expanded to include a detailed summary of existing research gaps and our motivation. This includes a discussion on the limitations of current methodologies and how our approach addresses these gaps.

4. Concern #4: The authors are suggested to use a separate related work discussion section.

o Author Response: We appreciate the suggestion to clearly explain the literature review section.

o Author Action: We have restructured and revised the related work in the "Related Work" section, providing a comprehensive review of existing studies and positioning our work within the broader research context.

5. Concern #5: A workflow figure can be added when describing the proposed methodology.

o Author Response: We agree that a workflow figure would aid in understanding the methodology.

o Author Action: A workflow figure has been added to illustrate the steps involved in our methodology, from data preprocessing to model evaluation. This figure helps clarify the process flow and integration of different models. Below figure is given below for methodology:

o 

6. Concern #6: It is not clear how to build the input and output for the CNN model.

o Author Response: We acknowledge the need for clearer explanations regarding the input and output configurations for the CNN model.

o Author Action: The manuscript has been revised to include a detailed explanation of the input and output structures for the CNN model, including the representation of trajectory data and the specific features used. This is now clearly mentioned in the methodology figure presented above and the mathematical detail is added to CNN model section 3.3.1. Figure 5 also show our model takes input parameter such as LAT,LON,SPEED and HEADING with the sliding window of size-3 and output the predicted trajectory such as LAT and LON.

o 

7. Concern #7: More baselines from previous studies are expected in the numerical experiments, instead of standard models.

o Author Response: We agree that including more comparative baselines would strengthen our evaluation.

o Author Action: As we have tested so many deep learning models but we have included the best performance models. As you can see the figure 7 and table 5 has been added to compare our proposed methodology with other existing models, and we achieved best accuracy as compared to other existing models. This provides a broader context for evaluating the performance of our proposed models and highlights the improvements achieved. The literature part is updated for previous baseline models and section 4.1 “Comparison with other existing methods” has been added for the comparative results with other baseline models.

o 

o 

Reviewer #2:

1. Concern #1: The English language usage needs refinement for clarity and coherence. Additionally, all abbreviations should be clearly defined upon first use and consistently used throughout the paper.

o Author Response: We recognize the importance of clear and precise language.

o Author Action: We have thoroughly revised the manuscript to improve language clarity and coherence. All abbreviations are now defined upon first use and consistently used throughout the text. The new table “Table 1. Abbreviations” is added for the abbreviations

2. Concern #2: Enhancing the literature review to include a broader range of related works would provide a stronger contextual foundation for the research.

o Author Response: We agree that a more comprehensive literature review would provide better context for our study.

o Author Action: The literature review has been expanded to include a broader range of related works, covering recent advancements and various methodologies in ship trajectory prediction and related fields. This addition strengthens the contextual foundation and highlights the novelty of our contributions.

3. Concern #3: The preprocessing technique details, such as the specific algorithm used for data cleaning and the fine-tuned parameters in the anomaly detection algorithm, need to be clarified.

o Author Response: We acknowledge the need for transparency in our preprocessing steps.

o Author Action: We have added detailed descriptions of the data cleaning algorithms and specific fine-tuned parameters used in the anomaly detection process. These details are included in the "Data Preprocessing" subsection, ensuring clarity and reproducibility. Preprocessing section is updated with some reference to other papers which describes the standard preprocessing techniques and a new algorithm “Algorithm 1: Enhanced AIS Data Preprocessing for Vessel Trajectory Extraction” is added which explains the whole process of preprocessing.

o 

4. Concern #4: The term "advanced CNN" needs to be clarified, including the specific advancements and differences from existing literature.

o Author Response: We understand the need to clearly define what constitutes our "advanced CNN."

o Author Action: We have updated the manuscript to clarify that the term "advanced CNN" refers to the novel preprocessing techniques we developed rather than changes to the CNN architecture itself. Specifically, the improvements in our methodology include:

1. Preprocessing Enhancements: We introduced advanced filtering techniques and a specialized data extraction approach that focuses on individual ships. This method improves data quality and captures unique behavioral patterns, addressing the limitations of conventional practices that aggregate data from multiple ships.

2. Individualized Data Extraction: Unlike traditional methods, our approach extracts and processes three months of data for each ship, which ensures privacy and allows for more precise trajectory predictions.

3. Expanded Evaluation Metrics: In addition to reducing Mean Absolute Error (MAE) and Mean Square Error (MSE), our model’s performance was also assessed using additional metrics such as Average Displacement Error (ADE), Final Displacement Error (FDE), and Non-Linear ADE (NL-ADE). These metrics provide a comprehensive evaluation of our model's effectiveness in managing trajectory prediction, particularly in capturing non-linear patterns and ensuring accurate final position predictions. The results show that our approach consistently performs well across these diverse metrics, demonstrating its robustness and accuracy.

5. Concern #5: There is a lack of transparency regarding whether the DNN model is adapted from previous work or newly designed.

o Author Response: We appreciate the need for transparency regarding the origins of our DNN model.

o Author Action: We have clarified that the Deep Neural Network (DNN) model employed in our research is utilized primarily for comparison purposes with the Convolutional Neural Network (CNN). The DNN model is used because it performs well with small datasets, such as the individual ship data we extracted. It is not a newly designed model but a standard approach included to benchmark the performance of CNN against other deep learning techniques. This clarification, along with a block diagram illustrating the DNN architecture, is provided in the revised manuscript to ensure transparency.

6. Concern #6: The paper lacks discussion about data preparation, the number of samples, and the training/testing split ratio.

o Author Response: We acknowledge the need for detailed information on data preparation.

o Author Action: For data preparation, we have addressed this by including a newly added methodology diagram, which provides a comprehensive overview of the preprocessing process. Specifically, we have added "Algorithm 1: Enhanced AIS Data Preprocessing for Vessel Trajectory Extraction," which outlines the entire preprocessing workflow. Additionally, in the results section, we have specified the training, validation, and testing sample ratios (70%, 15%, and 15%, respectively) to ensure transparency and reproducibility of our study.

---

## [Decision Letter · Decision Letter 1]

23 Aug 2024

PONE-D-24-15778R1Deep Learning Innovations in South Korean Maritime Navigation: Enhancing Vessel Trajectories Prediction with AIS DataPLOS ONE

Dear Dr. Kim,

Thank you for submitting your manuscript to PLOS ONE. After careful consideration, we feel that it has merit but does not fully meet PLOS ONE’s publication criteria as it currently stands. Therefore, we invite you to submit a revised version of the manuscript that addresses the points raised during the review process.

**ACADEMIC EDITOR: **Please define each abbreviation only once when it first appears in the text and use the abbreviation consistently thereafter. Also Please check the reviewer 2 comments regarding the formatting. 

We look forward to receiving your revised manuscript.

Kind regards,

Sushank Chaudhary, Ph.D

Academic Editor

PLOS ONE

Journal Requirements:

Reviewers' comments:

Reviewer's Responses to Questions

**Comments to the Author**

1. If the authors have adequately addressed your comments raised in a previous round of review and you feel that this manuscript is now acceptable for publication, you may indicate that here to bypass the “Comments to the Author” section, enter your conflict of interest statement in the “Confidential to Editor” section, and submit your "Accept" recommendation.

Reviewer #1: All comments have been addressed

Reviewer #2: All comments have been addressed

2. Is the manuscript technically sound, and do the data support the conclusions?

Reviewer #1: Yes

Reviewer #2: Yes

3. Has the statistical analysis been performed appropriately and rigorously? 

Reviewer #1: Yes

Reviewer #2: Yes

4. Have the authors made all data underlying the findings in their manuscript fully available?

Reviewer #1: Yes

Reviewer #2: Yes

5. Is the manuscript presented in an intelligible fashion and written in standard English?

Reviewer #1: Yes

Reviewer #2: Yes

6. Review Comments to the Author

Reviewer #1: no further comments for this paper. No additional comments for the author, including concerns about dual publication, research ethics, or publication ethics.

Reviewer #2: The authors have successfully addressed the majority of the previous comments. However, I have noticed that some abbreviations are repeated unnecessarily throughout the manuscript. It would be beneficial to define each abbreviation only once when it first appears in the text and use the abbreviation consistently thereafter. Additionally, the manuscript's formatting is inconsistent, particularly in the justification of the text. Ensuring that the text is uniformly justified will enhance the readability and overall presentation of the paper. These are minor issues that should be corrected before final acceptance.

7. PLOS authors have the option to publish the peer review history of their article (what does this mean?). If published, this will include your full peer review and any attached files.

Reviewer #1: No

Reviewer #2: No

---

## [Author Response · Author response to Decision Letter 1]

26 Aug 2024

Reviewer #2 Comments (Round-2)

Reviewer #2: 

Concern #1: The authors have successfully addressed the majority of the previous comments. However, I have noticed that some abbreviations are repeated unnecessarily throughout the manuscript. It would be beneficial to define each abbreviation only once when it first appears in the text and use the abbreviation consistently thereafter. Additionally, the manuscript's formatting is inconsistent, particularly in the justification of the text. Ensuring that the text is uniformly justified will enhance the readability and overall presentation of the paper. These are minor issues that should be corrected before final acceptance.

Author Response:

Thank you for your insightful feedback regarding the formatting and abbreviation usage in our manuscript. We have carefully reviewed the manuscript and made the necessary adjustments to ensure consistency in abbreviation usage, defining each abbreviation only once when it first appears in the text. Additionally, we have addressed the formatting issues by uniformly justifying the text throughout the document, enhancing the readability and overall presentation of our paper. We believe these revisions improve the clarity and quality of the manuscript.

ACADEMIC EDITOR: 

Please define each abbreviation only once when it first appears in the text and use the abbreviation consistently thereafter. Also, please check the reviewer 2 comments regarding the formatting.

Author Response:

We appreciate the Academic Editor's feedback and have thoroughly reviewed the manuscript to ensure that all abbreviations are defined at their first mention and are used consistently throughout the text. We understand the importance of maintaining consistency in abbreviation usage for clarity and have implemented the necessary changes.

In response to Reviewer 2's comments regarding the formatting, we have also ensured that the text is uniformly justified throughout the manuscript. We believe these revisions will significantly enhance the readability and presentation of our paper.

---

## [Editor Report · Decision Letter 2]

30 Aug 2024

Deep Learning Innovations in South Korean Maritime Navigation: Enhancing Vessel Trajectories Prediction with AIS Data

PONE-D-24-15778R2

Dear Dr. Kim,

We’re pleased to inform you that your manuscript has been judged scientifically suitable for publication and will be formally accepted for publication once it meets all outstanding technical requirements.

Kind regards,

Sushank Chaudhary, Ph.D

Academic Editor

PLOS ONE
---

## [Editor Report · Acceptance letter]

14 Oct 2024

PONE-D-24-15778R2 

PLOS ONE

Dear Dr. Kim, 

I'm pleased to inform you that your manuscript has been deemed suitable for publication in PLOS ONE. Congratulations! Your manuscript is now being handed over to our production team.

Kind regards, 

on behalf of

Prof. Sushank Chaudhary 

Academic Editor

PLOS ONE